# Dissecting the Many Faces of Frontotemporal Dementia: An Imaging Perspective

**DOI:** 10.3390/ijms232112867

**Published:** 2022-10-25

**Authors:** Marta Pengo, Enrico Premi, Barbara Borroni

**Affiliations:** 1Department of Molecular and Translational Medicine, University of Brescia, 25123 Brescia, Italy; 2Stroke Unit, Department of Neurological and Vision Sciences, ASST Spedali Civili, 25123 Brescia, Italy; 3Neurology Unit, Department of Neurological and Vision Sciences, ASST Spedali Civili, 25123 Brescia, Italy; 4Centre for Neurodegenerative Disorders, Department of Clinical and Experimental Sciences, University of Brescia, 25123 Brescia, Italy

**Keywords:** frontotemporal dementia, neuroimaging, biomarkers, MRI, PET, neurodegeneration

## Abstract

Frontotemporal dementia (FTD) is a heterogeneous clinical and neuropathological disorder characterized by behavioral abnormalities, executive dysfunctions and language deficits. FTD encompasses a wide range of different pathological entities, associated with the accumulation of proteins, such as tau and TPD-43. A family history of dementia is found in one third of cases, and several genes causing autosomal dominant inherited disease have been identified. The clinical symptoms are preceded by a prodromal phase, which has been mainly studied in cases carrying pathogenetic mutations. New experimental strategies are emerging, in both prodromal and clinical settings, and outcome markers are needed to test their efficacy. In this complex context, in the last few years, advanced neuroimaging techniques have allowed a better characterization of FTD, supporting clinical diagnosis, improving the comprehension of genetic heterogeneity and the earliest stages of the disease, contributing to a more detailed classification of underlying proteinopathies, and developing new outcome markers on clinical grounds. In this review, we briefly discuss the contribution of brain imaging and the most recent techniques in deciphering the different aspects of FTD.

## 1. Introduction

Frontotemporal dementia (FTD) is a common cause of early-onset dementia [1,2], with devasting psychological and social implications for both patients and families.

FTD encompasses a heterogeneous group of neurodegenerative disorders with a wide range of clinical, genetic and neuropathological features [2,3,4,5]. The careful characterization of clinical features of the behavioral variant frontotemporal dementia (bvFTD) [3], the agrammatic or the semantic variant of primary progressive aphasia (avPPA and svPPA) [4], and the spectrum of frontotemporal lobar degeneration (FTLD) with extrapyramidal symptoms, such as corticobasal syndrome (CBS) and progressive supranuclear palsy (PSP), has enabled a better understanding of the heterogeneity of FTLD phenotypes [6].

Along with the heterogeneity of clinical presentations, a complex neuropathology is associated with the disease, characterized by FLTD-Tau, FTLD-TDP and FTLD-FET with respect to the deposition of the tau protein, TAR DNA Binding Protein 43 (TDP-43) and FET family proteins, respectively [7,8].

In one third of cases, FTD is associated with an autosomal dominant inherited mutation in one of three main genes: microtubule-associated protein tau (*MAPT*), progranulin (*GRN*), and chromosome 9 open reading frame 72 (*C9orf72*) [9].

This heterogeneity, as well the lack of a clear-cut relationship between clinical phenotypes, genetic traits and neuropathological features, represents the main obstacle hampering the development of a unifying disease model and, as consequence, disease-modifying strategies of intervention.

In the last few years, advanced neuroimaging techniques have gone beyond the mere neuroanatomical description of frontotemporal atrophy or hypometabolism in FTD patients, and have helped in increasing the diagnostic accuracy, in disentangling the features associated with monogenic disease, in describing the earliest changes occurring in the prodromal phases, and in forecasting disease progression [10]. Moreover, new positron emission tomography tracers provide key information to define the underlying neuropathology [11,12,13,14,15]. These neuroimaging developments have contributed to the exploration of FTD pathogenesis and to the identification of novel potential biomarkers. Since at present there is a lack of disease-modifying therapies in FTD and treatment relies on symptomatic interventions [16], brain imaging biomarkers might be crucial in facilitating the recruitment of patients in clinical trials.

In the present review, we discuss the most important brain imaging candidates that help decipher the different aspects of FTD, and suggest an approach to further improve our knowledge in FTD-related imaging (Figure 1).

## 2. Neuroimaging and FTD Phenotypes

The different FTD syndromes are characterized by an early involvement of the insula and the anterior cingulate, which are part of the salience network [17,18,19,20]. Besides this common feature, structural brain magnetic resonance imaging (MRI) and brain positron emission tomography with [F18] fluorodeoxyglucose (FDG-PET) show characteristic and distinctive neuroimaging patterns, which can help accurately discriminate between different FTD phenotypes and are routinely used in clinical practice [3,4,21,22,23,24,25,26,27]. By means of volumetric T1-weighted MRI, it is possible to detect changes in grey matter (GM) structure, determine volumes of specific regions of interest and the rate of atrophy. In addition, postprocessing techniques can be applied to structural images, such as investigations of changes at the voxel level through voxel-based morphometry or measurement of cortical thickness. On the other hand, FDG-PET allows the identification of alterations in the brain metabolism that might precede GM atrophy [28,29].

Conversely, advanced MRI methods, such as diffusion tensor imaging (DTI), functional MRI (fMRI) and arterial spin labeling (ASL), are currently used in research and might be more sensitive in the earliest phases of the disease [30,31,32,33,34,35]. DTI explores microstructural white matter (WM) alterations that anticipate GM loss in FTD [36,37], whereas fMRI is a technique sensitive to changes in functional brain connectivity. ASL reveals alterations in brain perfusion that correlate very well with metabolism measured with FDG-PET [38,39]. These MRI techniques have the advantages of being safe, non-invasive, repeatable, and able to be combined in a single session, and not involving radiation exposure. However, they have been investigated at a group level, and currently are not applicable in clinical practice at a single-patient level. Moreover, fMRI studies are limited by a wide variation in analytical methods used, such as independent component analysis, seed-based or region-of-interest-based approaches.

Atrophy in bvFTD involves the frontal and temporal lobes, the insula and the anterior cingulate cortex, reflecting the distribution of Von Economo neurons [22,23,40,41] (Figure 2). The pattern of atrophy in bvFTD is usually asymmetric, predominantly involving the right hemisphere, and is associated with the core neuropsychiatric features, including disinhibition, apathy, loss of empathy and binge eating [42,43,44,45,46,47]. The degeneration of many subcortical structures is also observed in bvFTD, in the amygdala, hippocampus, basal ganglia and thalamus [48].

Brain FDG-PET shows areas of hypometabolism that reflect atrophic regions, but might be more sensitive than structural images in the initial stages of the disease [28,29]. Low glucose metabolism is observed in comparable brain structures, mainly in the orbitofrontal, dorsolateral and medial prefrontal cortices, anterior temporal poles and basal ganglia [39,49,50].

The distribution of atrophy and hypometabolism helps in differentiating FTD from Alzheimer’s disease (AD) [25,29,51,52,53,54,55,56,57,58].

The avPPA is mainly identified by left-sided frontal and insula involvement, both at structural MRI and brain FDG-PET [24,26,27,59,60], while the svPPA presents with asymmetrical, typically left-sided, anteroinferior temporal lobe atrophy and hypometabolism [24,27] (Figure 2). A minority of patients can exhibit right-predominant patterns of atrophy/hypometabolism affecting the temporal lobe and may present clinically with prosopagnosia, memory impairment and behavioral changes [61,62,63,64].

Distinct patterns of metabolic abnormalities in primary progressive aphasia (PPA) are important not only for the differential diagnosis of the different syndromes, but also to predict progression to specific dementia subtypes [27]. In avPPA, hypometabolism involving parietal, subcortical and brainstem structures was associated with progression to CBS or to PSP. svPPA showing extended bilateral hypometabolism progressed to bvFTD over time.

## 3. Neuroimaging and Neuropathology

One of the main goals of the current literature in FTD is to develop reliable imaging markers able to predict in vivo neuropathological hallmarks, namely tau or TDP-43 inclusions. Identifying biomarkers of misfolded proteins is extremely relevant for a precision medicine approach in future clinical trials. This would be key especially in non-monogenic cases for which neuropathology is still unpredictable.

To date, at the single-subject level, no brain MRI approaches hold the premise to identify neuropathology in FTD patients. Moreover, whereas the pattern of GM atrophy does not differ between FTLD-tau and FTLD-TDP [65], it has been described that FTLD-tau had significantly more WM degeneration in post mortem studies compared to FTLD-TDP [66].

Conversely, the field of PET imaging with tracers targeting different proteins has exploded in recent years. This technique holds the tremendous potential to define not only the underlying neuropathology of neurodegenerative diseases, but also the pattern of distribution of unfolded proteins. Thus, this will be important also to investigate disease pathogenesis for early and differential diagnosis and to monitor disease progression.

Tau PET first-generation tracers have led to inconclusive results because of the lack of specificity, subcortical WM uptake, and variable affinity for different tau isoforms [11,12,13,14,15]. These limitations have prompted the development of the second generation of tau PET tracers. It has been demonstrated that these new radiotracer have a high affinity for tau neurofibrillary tangles, the hallmark of AD pathology [67,68]. In the same view, tau PET shows a good sensitivity in carriers of *MAPT* mutations that are more likely to cause an AD-like tau pathology [11,14,69,70]. However, these tracers bind only weakly, if at all, to 3-repeat tau in the Pick bodies of FTD and the 4-repeat linear tangles in dementias associated with PSP and CBS [68,71,72,73,74,75]. Moreover, although different studies demonstrated in vivo increased tau accumulation in the midbrain in PSP, discrepancies with autopsy studies and considerable overlap with healthy controls (HC) underline the ongoing need for further investigations in this field [15,74,76].

No specific TDP-43 tracers are available yet. However, in different series of patients with svPPA, a disease typically characterized by TDP-43 pathology, tau PET signal was unexpectedly elevated with spatial distribution mirroring areas of atrophy. These results raise concerns about the lack of specificity of tau tracers, suggesting a possible off-target biding to non-tau molecules [77,78,79,80]. PET ligands developed to bind tau neurofibrillary tangles in AD showed increase uptake also in bvFTD due to hexanucleotide repeat expansion in *C9orf72*, associated with TDP-43 deposition [12].

To conclude, it is clear that the field of PET imaging is extremely promising and progressing very rapidly. Nevertheless, further research is warranted in the spectrum of FTLD to clarify the aforementioned ambiguities. Furthermore, confirmation of in vivo findings with autopsy studies will be necessary for the validation of tau tracers in this field.

## 4. Neuroimaging and Genetics

In recent years, the identification of new causative genes associated with FTD has represented a giant step forward to characterize the heterogeneity of the disorder, at the clinical, molecular and imaging levels. Moreover, exploring genetic FTD is crucial since it represents the ideal target population for the development of disease-modifying therapies and allows to unravel the prodromal disease stages in at-risk subjects [81,82,83,84,85,86]. Thus, defining biomarkers in the preclinical as well as clinical stages represents a priority in order to stratify patients for clinical trials and to assess the efficacy of therapeutic interventions in this population.

Each of the most common genetic groups, namely *GRN*, *MAPT* and *C9orf72* mutations, display a differential and characteristic pattern of cortical atrophy [87] with early changes appearing during the prodromal disease stages, up to 20 years before phenoconversion [82,88]. These results were obtained by well-established international networks, such as the European- and Canadian-based Genetic Frontotemporal dementia Initiative (GENFI, www.genfi.org), the US-based ARTFL/LEFFTDS, and the Australian DINAD, which have collected cross-sectional and longitudinal data of FTD patients with monogenic disease [9,82]. In addition, the recently established consortia in Latin America (ReDLat) and New Zealand (FTDGeNZ) will be able to further elucidate the natural history of the disease [89,90].

Symptomatic stages of the disease can be investigated accurately with structural MRI. Comparable atrophy for all three mutation groups was observed in a network involving the insula, orbitofrontal lobe and anterior cingulate. Besides these areas, each mutation group develops a characteristic pattern of cortical atrophy [88]. *C9orf72* mutation carriers present atrophy symmetrically, involving dorsolateral, medial and orbitofrontal lobes. Anterior temporal lobes, thalamus, parietal and occipital lobes and cerebellum are also affected [23,82,88,91,92,93,94,95,96]. Cases of patients with mild, slowly progressive or even severe dementia with minimal or no atrophy have been reported [91,97,98]. Interestingly, a comparable pattern of functional network alterations despite various atrophy patterns have been described in *C9orf72* expansion carriers. In particular, they are characterized by reduced connectivity in the salience network and sensorimotor networks, whereas default mode network connectivity is similar to HC, unlike sporadic FTD [99].

*GRN* mutations carriers display a characteristic striking asymmetrical atrophy involving frontotemporal but also parietal cortices [23,82,88,95,100,101,102]. Both left- and right-sided predominant atrophy can be observed, even in the same family. *GRN* mutation carriers can present WM hyperintensities, even in the pre-symptomatic stages, which might be due to microglial activation and microglial dystrophy [103,104,105,106,107].

The distribution of atrophy in patients with *MAPT* mutation symmetrically involves the anterior and mesial temporal lobes, whereas orbitofrontal, lateral prefrontal and parietal regions are less altered [23,82,88,95,100]. A differential involvement of the temporal lobe has been described according to the diverse mutations in the *MAPT* gene [100].

The rate of atrophy varies in the different forms of genetic FTD. It is faster in those with *GRN* mutations and slower in *MAPT* mutation carriers. *C9orf72* mutation carriers show the greatest heterogeneity in the progression of brain atrophy [108,109]. In *GRN* mutation carriers, after clinical onset, the rate of atrophy is greater in the temporal cortex and becomes more asymmetrical in the following stages of the disease [82].

Besides the most frequent mutations, more rare pathogenic mutations are associated with genetic FTD. At present, the neuroimaging findings in these patients are described in case reports or in a limited number of cases, and future international studies with larger cohorts are warranted in this field.

## 5. Neuroimaging and Prodromal Stages

Prodromal FTD may be defined as the presence of mild cognitive and/or behavioral changes without a significant impact on functional independence. The label of mild cognitive and/or behavioral and/or motor impairment (MCBMI) was recently proposed in a consensus paper with the aim of capturing the complexity of the clinical presentation in this disease stage [110].

In this context, the genetic forms have provided a privileged point of view for investigating the prodromal phases, assessing brain changes in at-risk mutation carriers.

Individuals with FTD-associated mutations develop GM atrophy and hypometabolism at least 10 years before symptom onset, whereas WM abnormalities and functional connectivity alterations are seen even earlier, supporting the hypothesis of FTD as a network-based disease. Therefore, DTI and fMRI seem to be the most promising techniques to explore pre-symptomatic stages, with a greater sensitivity than structural MRI [31,32,33,111,112,113].

Traditionally, brain networks have been regarded as static over time. However, a recently introduced evolution of the brain connectome, the so-called chronnectome, allows to capture the dynamic functioning of the brain across time [114,115,116]. This approach has been demonstrated to be even more sensitive than traditional resting state fMRI approaches in pre-symptomatic phases [117,118]. ASL MRI is also a promising non-invasive imaging biomarker in pre-symptomatic carriers. Indeed, cerebral blood flow differences appeared earlier than 10 years before the expected onset in key FTD regions in the GENFI cohort [35].

The different genetic groups, such as *GRN, C9orf72* and MAPT mutations, are characterized by variability in both timing and location of early GM and WM changes.

In *C9orf72* mutation carriers GM and WM alterations appear very early, up to 30 years before symptoms onset; *GRN* mutation carriers show no or only minimal atrophy in pre-symptomatic stages, thus representing the most challenging group to investigate in these disease stages [30,32,33,34,82,88,99,112,119,120,121,122]. Conversely, studies in pre-symptomatic *MAPT* mutation carriers showed contrasting results, probably because of the differences in cohort size and subject heterogeneity [113,119,120,123]; the largest study at present found early volume loss in hippocampus and amygdala at 15 years before expected onset [82].

## 6. Neuroimaging & Disease Progression

Conventional neuroimaging is routinely used for clinical diagnosis [124] and might also provide prognostic information [125,126]. Since FTD is characterized by great heterogeneity in clinical course, identifying imaging biomarkers of disease progression would be crucial also for inclusion in clinical trials.

The pattern of brain atrophy in FTD carries also prognostic information, as the degree of atrophy in the anterior cingulate and motor cortex predicted a faster disease progression [127], while diffuse brain atrophy was related to a worse prognosis than focal atrophy [128]. Functional brain imaging might also aid in defining prognosis. Hypoperfusion in the orbitomesial frontal cortex associated with the “pseudomanic behavior”, characterized by disinhibition and abnormal social conduct, predicted a worse prognosis [129]. Moreover, hypoperfusion in the right orbitomesial frontal cortex and in the brainstem was associated with decreased survival [130,131]. Accordingly, different patterns of metabolic abnormalities in PPA are important to predict progression to specific dementia subtypes [27].

Along the disease course, the most suitable neuroimaging biomarker and the regions to evaluate could change. In this context, multivariate statistical approaches, like the multi-voxel pattern analysis (MVPA), might be useful for establishing the most accurate biomarker for clinical trials. By means of MVPA, it was demonstrated that in patients carrying *GRN* mutation, the most predictive measures were structural alterations, whereas in pre-symptomatic carriers, the best predictors markers were functional abnormalities, in particular the local connectivity measure (fALFF) [112].

Biological modulators and environmental factors, like education, have been proposed to contribute to heterogeneity in disease progression both in sporadic and genetic forms [132,133,134,135,136]. Neuroimaging might be of crucial relevance in evaluating the influence of these factors. Actually, disease modifiers can modulate brain atrophy [137] and brain connectivity [138] even in pre-symptomatic phases [139,140]. As a consequence, taking into account their effect of early brain damage is mandatory to stratify patients at risk of dementia, develop new therapies, and eventually monitor the efficacy of treatments.

Despite the relevance of neuroimaging as biomarker in FTD, at present, its potential has been investigated mainly at the group level to discriminate patients’ groups from each other or from HC. However, this poses difficulties in clinical practice, since diagnostic and prognostic information at a single-subject level is essential. Recently developed tools might help to overcome this limitation. The preGRN-MRI tool was able to predict the expected MRI atrophy at follow-up using baseline MRI measures in pre-symptomatic *GRN* mutation carriers with good accuracy [141]. Therefore, this tool can be helpful in clinical trials since deviation of the cortical thickness from the expected model might be a marker of treatment efficacy.

Moreover, machine-learning techniques are able to stratify in vivo disease subtypes and stages, and recent studies demonstrated their reliability for dealing with the extreme heterogeneity of different neurodegenerative diseases. They therefore appear to be promising approaches in precision medicine in order to stratify patients at very early disease stages. Subtype and stage inference (SuStaIn) is a computational approach that unravels phenotypic heterogeneity to distinguish patients’ subgroups with a similar pattern of disease progression [142]. In genetic FTD, using structural T1-weighted imaging, it was able to identify genotypes from imaging alone. Moreover, it could even reveal within-genotype heterogeneity, namely, different subgroups in *C9orf72* and *MAPT* mutation carriers, characterized by diverse disease trajectories [142,143]. The contrastive trajectory inference (cTI) is another recently developed machine learning algorithm for staging and subtyping disease. It has been already applied for different neurodegenerative diseases, such as AD and Huntington’s disease [144]. A recent work explored its reliability also in genetic FTD [145]. Indeed, cTI could stage the disease in a heterogeneous cohort of genetic FTD, both pre-symptomatic and symptomatic, using only a combination of different neuroimaging modalities without clinical information. Therefore, machine learning appears to be a promising approach to follow disease progression and monitor treatment efficacy in future clinical trials. A further development might be the combination with non-imaging biomarkers, e.g., neurofilament light chain, for optimal disease staging.

## 7. Conclusions and Future Perspectives

Neuroimaging appears to be a key biomarker in FTD. Whereas some techniques, such as structural MRI and FDG-PET, are routinely used in clinical practice mainly for diagnostic purposes, novel emerging techniques are under development with different aims. PET with specific tracers and advanced neuroimaging approaches, such as fMRI and DTI, will be essential to define the underlying neuropathology and investigate pre-symptomatic disease stages, respectively. Machine learning approaches will be crucial for early diagnosis, to evaluate disease progression and stratify patients for future clinical trials, and eventually also to combine neuroimaging with non-imaging biomarkers. All these advancements in neuroimaging research will be essential to develop and monitor new therapies in a pathology which is still an orphan of disease-modifying treatments.

## Figures and Tables

**Figure 1 ijms-23-12867-f001:**
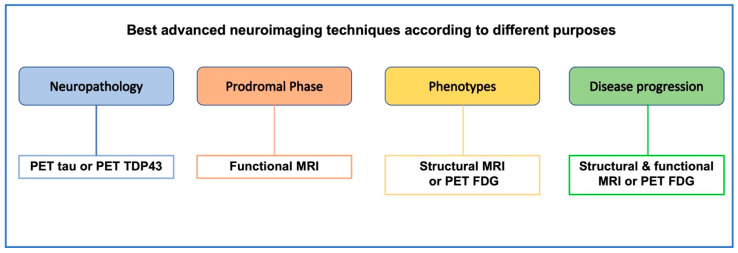
Proposed advanced neuroimaging techniques according to different aims. Advanced neuroimaging allows to investigate different aspects of FTD, from defining neuropathology to early and differential diagnosis and disease progression. Different techniques are suitable for each of these aims.

**Figure 2 ijms-23-12867-f002:**
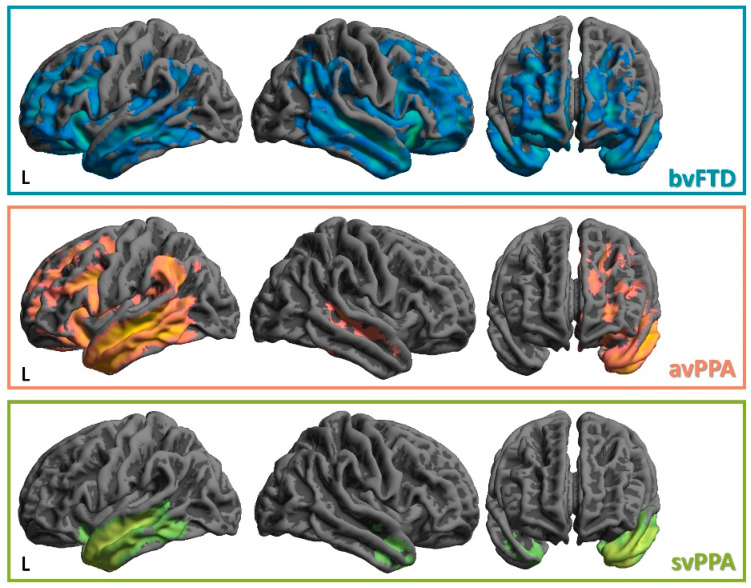
Representative VBM analysis in FTLD. From our historical cohort of FTLD patients we performed a structural MRI analysis (voxel-based morphometry) to demonstrate the different patterns of gray matter atrophy. To this aim, we considered the following cohort of subjects (80 healthy controls (63.1 ± 7.9 years, 75% females); 122 bvFTD (64.8 ± 7.9 years, 38% females): 68 avPPA (65.6 ± 8.4 years, 66.2% females); 30 svPPA (63.6 ± 8.4 years, 56.7% females). Age and gender are considered nuisance variables in the statistical model. The findings (patient < healthy controls for each group) are superimposed on a 3D MRI template; clusters surviving a statistical threshold of *p* < 0.05 FWE whole-brain correction for multiple comparisons are reported. VBM—voxel-based morphometry; MRI—magnetic resonance imaging; FWE—family-wise error; L—left.

## Data Availability

All study data are available from the corresponding author, upon reasonable request.

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
