# Peer review of "Dissecting the Many Faces of Frontotemporal Dementia: An Imaging Perspective"

_ijms, 2022, doi:10.3390/ijms232112867_

Round 1

Reviewer 1 Report

This review summarizes the latest advances in neuroimaging techniques in frontotemporal dementia and provides an important reference for scholars in this field. I recommend this article for publication in the International Journal of Molecular Sciences with some positive comments as follows:

1 Different neuroimaging techniques need to be classified. Summarize the advantages and disadvantages of each technique.

2. The harm of frontotemporal dementia and its diagnosis and treatment status are introduced too little.

3. Add some beautiful illustrations appropriately to facilitate readers reading and understanding. 4. The font size should not be unified. The reference format needs to be unified.

Author Response

This review summarizes the latest advances in neuroimaging techniques in frontotemporal dementia and provides an important reference for scholars in this field. I recommend this article for publication in the International Journal of Molecular Sciences with some positive comments as follows:

1 Different neuroimaging techniques need to be classified. Summarize the advantages and disadvantages of each technique.

Answer: we thank the reviewer for this comment, which gives us the opportunity to clarify the different techniques. We have modified the manuscript in order to illustrate advantages and disadvantages of each of them (page 5, lines 72-77 and 80-88).

  1. The harm of frontotemporal dementia and its diagnosis and treatment status are introduced too little.

Answer: we have modified the introduction to address these points (pages 3, lines 35-36 and page 4, lines 58-62).

  1. Add some beautiful illustrations appropriately to facilitate readers reading and understanding.

Answer: we have included illustrations to visualize some key findings in the field of FTD neuroimaging (Figure 2).

  1. The font size should not be unified. The reference format needs to be unified.

Answer: we have modified the manuscript in accordance to these suggestions.

Reviewer 2 Report

This is an interesting review of the current state of diagnosis and treatment of frontotemporal dementia. The authors appear to be leaders in this area of clinical neuroscience. The field itself appears about to explode in terms of clinical and diagnostic approaches, especially in the realms of genetic markers and imaging, that will prove beneficial to patients. In this regard, current results seem quite disparate: one wonders if establishing an international database of dementia observations- similar to the genetic consortiums, would prove useful to the field. 

The references are appropriate and well-done.

This reviewer recommends a thorough editing by one native to written English as there are a number of mismatched pronouns, ambiguities and awkward phrases.

Overall: excellent contribution.

Author Response

This is an interesting review of the current state of diagnosis and treatment of frontotemporal dementia. The authors appear to be leaders in this area of clinical neuroscience. The field itself appears about to explode in terms of clinical and diagnostic approaches, especially in the realms of genetic markers and imaging, that will prove beneficial to patients. In this regard, current results seem quite disparate: one wonders if establishing an international database of dementia observations- similar to the genetic consortiums, would prove useful to the field. 

The references are appropriate and well-done.

This reviewer recommends a thorough editing by one native to written English as there are a number of mismatched pronouns, ambiguities and awkward phrases.

Overall: excellent contribution.

Answer: the manuscript has been revised by a native English-speaker to improve this aspect.  

Round 2

Reviewer 1 Report

The revised manuscript has answered my question very well. I recommend it for publication in the International Journal of Molecular Sciences.